# National Assessment of Opportunities for Improvement in Preventable Trauma Deaths: A Mixed-Methods Study

**DOI:** 10.3390/healthcare11162291

**Published:** 2023-08-14

**Authors:** Junsik Kwon, Myeonggyun Lee, Kyoungwon Jung

**Affiliations:** 1Division of Trauma Surgery, Department of Surgery, Ajou University School of Medicine, Suwon 16499, Republic of Korea; aquaestel@aumc.ac.kr; 2Division of Biostatistics, Department of Population Health, New York University Grossman School of Medicine, New York, NY 10016, USA; ml5977@nyu.edu

**Keywords:** mortality, patient transfer, quality improvement, treatment outcome, wounds and injuries

## Abstract

Trauma is a significant public health issue worldwide, particularly affecting economically active age groups. Quality management of trauma care at the national level is crucial to improve outcomes of major trauma. In Korea, a biennial nationwide survey on preventable trauma death rate is conducted. Based on the survey results, we analyzed opportunities for improving the trauma treatment process. Expert panels reviewed records of 8282 and 8482 trauma-related deaths in 2017 and 2019, respectively, identifying 258 and 160 cases in each year as preventable deaths. Opportunities for improvement were categorized into prehospital, interhospital, and hospital stages. Hemorrhage was the primary cause of death, followed by sepsis/multiorgan failure and central nervous system injury. Delayed hemostatic procedures and transfusions were common areas for improvement in hospital stage. Interhospital transfers experienced significant delays in arrival time. This study emphasizes the need to enhance trauma care by refining treatment techniques, centralizing patients in specialized facilities, and implementing comprehensive reviews and performance improvements throughout the patient transfer system. The findings offer valuable insights for addressing trauma care improvement from both clinical and systemic perspectives.

## 1. Introduction

Trauma is a major cause of death among individuals aged <40 years in Korea and worldwide. Unlike diseases such as cardiovascular disorders or malignant tumors, injury is considered an important public health issue in most countries due to its impact on mortality and disability in economically active age groups [1,2,3,4]. Leading nations in trauma care have implemented strict national-level trauma management quality to reduce the significant socio-economic losses caused by trauma [5]. Expert panel reviews of trauma deaths are the most representative quality management approaches in trauma care, aimed at evaluating the quality of trauma treatment and identify potentially modifiable treatment practices [6]. Although this method has inherent limitations from relying on subjective assessments by the panel, it remains crucial for the intuitive assessment of a nation’s capacity in the field of trauma care. The preventable trauma death rate (PTDR) is defined as the proportion of trauma-related deaths that could have been prevented if the patient had been appropriately transferred to a suitable hospital within a suitable timeframe and received appropriate treatment, as determined through expert reviews of such trauma deaths [4]. In Korea, periodic surveys on PTDR have been conducted since the late 1990s. The PTDR in Korea was initially very high at 40.5% in 1997; however, it gradually decreased as the national trauma care system was established [7,8,9,10,11,12]. According to the most recent national survey conducted in 2019 and targeting trauma-related deaths, the PTDR was found to be 15.7%, with a decrease of 4.2% compared to that reported in the previous survey conducted in 2017 [13]. Although ongoing related studies may raise concerns about the reliability of objective numerical values [14,15], panels agree that identifying the opportunities for improvements (OFIs) discovered during the treatment process is the most powerful method [15]. Identifying and categorizing OFIs in a step-by-step manner provides guidance on specific efforts to reduce PTDR [16]. This study aimed to identify and analyze OFIs at the prehospital, interhospital, and hospital stages through expert panel reviews of trauma deaths in 2017 and 2019, examining their nature and proportions. It is expected that such research will be useful in establishing priorities and strategies for the development and improvement of the Korean trauma system.

## 2. Materials and Methods

### 2.1. Study Design—Population and Data Sampling

This study utilized data from patients who visited hospitals and subsequently died due to trauma in 2017 and 2019. Patients who experienced trauma were defined as those with one or more S or T codes according to the Korean Standard Classification of Diseases (KCD), seventh edition [17]. KCD is a systematic classification of diseases and deaths in Korea. It has been in use since 1952 to standardize criteria for compiling statistics on public health and medical phenomena. Based on the International Statistical Classification of Diseases and Related Health Problems (ICD), KCD provides a standardized framework for data analysis. Within the KCD system, the disease classification codes S and T specifically identify health problems resulting from trauma. The national emergency department information system (NEDIS) was used as a sampling framework for retrieving mortality statistics for patients who experienced trauma [18]. NEDIS is a national database that includes clinical and administrative data of all patients visiting emergency departments. In 2017 and 2019, there were 8282 and 8482 trauma-related deaths, respectively, as aggregated from emergency medical institutions nationwide. To obtain an unbiased sample that reflects the characteristics of the population, a stratified two-stage cluster sampling method with stratification and dual-stage clustering was employed. For the first stage of stratification, variables such as region, hospital type (regional trauma center, regional emergency medical center, and local emergency medical institution), and the number of deaths (≥100, 50–99, 30–49, 10–29, and <10) were used. For the second stage of stratification, variables including time of death and patient age were used (Figure 1).

### 2.2. Research Participants—Data Collection Method and Types

Based on the stratified two-stage cluster random sampling, the research team requested medical records of the selected patients from the hospitals where they were treated; records of a sample of 1862 and 1692 patients were retrieved for the 2017 and 2019 cohorts, respectively. The requested medical records were submitted to the Ministry of Health and Welfare through the hospitals in accordance with the Emergency Medical Services Act and the Medical Act. Important imaging examination results were stored on compact disks and submitted with their interpretations. The requested medical records included initial emergency room visit records, progress notes, nursing charts, official imaging interpretation sheets, blood test results, and discharge records. Additionally, ambulance records were collected for the analysis of prehospital stage OFIs.

### 2.3. Research Instrument—Multidisciplinary Case Review and Preventable Trauma Death Rate

To review medical records, the research team developed a structured review form based on the data sheet proposed by the World Health Organization guidelines [4] (Appendix A). A total of 32 preliminary surveyors, mainly trauma coordinators working at regional trauma centers, organized and summarized the medical records before the expert panel review. All surveyors had completed training on case reviews provided by the National Medical Center. To identify OFIs in reviewed cases and assess preventability, a panel of trauma care specialists consisting of 25 trauma physicians working mainly at regional trauma centers was selected. After independently reviewing each case, they participated in discussions in groups composed of two general surgeons, one thoracic surgeon, one neurosurgeon, and one emergency medicine physician. The panel members assessed the preventability of trauma deaths for 1251 cases in 2017 and 1208 cases in 2019, excluding cases that were determined not to be trauma-related deaths. The final calculated PTDR was 19.9% in 2017 and 15.7% in 2019. It should be noted that the PTDR was calculated using weighted estimates, rather than actual numbers, and may differ from calculations using actual numbers.

### 2.4. Analysis—Qualitative Analysis of Preventable Trauma Death Cases

After determining the preventability for each patient with a trauma-related death, the panel identified OFIs in the patient’s treatment process. The identified OFIs were categorized into prehospital, interhospital, and hospital stages. The preventable trauma death cases were classified according to the primary causes of death, and the results were compared with those of a previous study conducted on deceased individuals in 2017 [8]. The performance and timing of major procedures such as transfusion, hemostasis surgeries, craniectomy, and intubation, which were performed for patients who died from bleeding or severe brain injury, were examined.

### 2.5. Ethics Statement

This study was approved by the Ajou University Institutional Review Board (AJIRB-MED-EXP-20-473). Due to the observational nature of the study, the board waived the requirement for informed consent.

## 3. Results

### 3.1. OFIs in Preventable Cases by Phase of Care

A total of 258 and 160 preventable trauma death cases were identified in 2017 and 2019, respectively. The number of OFIs in these cases decreased from 306 in 2017 to 212 in 2019. In 90% of the cases from 2019, OFIs were identified in the hospital stage, while OFIs in the interhospital stage and prehospital stage were identified in 24.4% and 18.1% of cases, respectively. In cases from 2017, although a similar number of OFIs were identified in the prehospital stage compared to that in 2019, there were 19 more OFIs identified in the interhospital stage and 76 more OFIs identified in the hospital stage (Table 1).

### 3.2. Time from the Scene to the Destination Hospital

Among the investigated death cases, the time from injury to arrival at the destination hospital was available for 1173 cases in 2017 and 913 cases in 2019. In 2019, when there was at least one interhospital transfer (196 cases), the median time was 3 h and 22 min, which was significantly delayed compared to a median time of 35 min for cases with direct arrival (716 cases). The 2017 results showed similar median times of 3 h and 2 min for all cases and 33 min for cases with direct arrival, with no statistical difference compared to results from 2019. When considering only cases classified as preventable trauma deaths, in 2019, the median time for cases with at least one interhospital transfer (47 cases) was 4 h and 55 min, indicating a delay compared to the median of 40 min for cases with direct arrival (97 cases). Specifically, among the cases with OFIs identified in the interhospital stage (11 cases), the median time was 5 h and 44 min. There was no statistical difference compared to the results from 2017 (Table 2).

### 3.3. Classification by Cause of Death

Among the 160 preventable trauma death cases, hemorrhage was the leading cause of death in 56 cases (35.0%), followed by sepsis or multiorgan failure and central nervous system injury. Additionally, in some cases the death was due to respiratory and cardiac issues, and in seven cases the cause of death was undetermined. In cases with preventable trauma deaths in 2017, although the causes of death showed a generally similar distribution, there was a difference in which central nervous system injury accounted for more deaths than sepsis/multiorgan failure (Table 3).

### 3.4. OFIs by Treatment Stage for Patients Who Died from Hemorrhage

For cases where hemorrhage was the leading cause of death, the identified OFIs were categorized based on the stage of treatment. The number of OFIs in the hospital stage was 157, which was higher compared to that in the prehospital and interhospital stages. Specifically, “delay in hemostatic procedures” was the most commonly identified OFI (46 cases, 29.3%), followed by “delay in procedures other than hemostasis” (42 cases, 26.8%) and “delay in transfusion”. These were also the most frequently identified OFIs in the hospital stage in 2017. In the interhospital transfer stage, OFIs such as “delay in transfer”, “delay in transfusion”, and “delay in hemostatic procedures” were frequently identified. The 2017 study showed a difference in the frequency of OFIs, with “delay in procedures other than hemostasis”, “delay in transfusion”, and “delay in transfer” being the most commonly identified OFIs. In the prehospital stage, “inadequate hospital selection” was the most frequently identified OFI, followed by “fluid resuscitation” and “delay in transfer”. The results from the 2017 survey showed that “fluid resuscitation”, “inadequate hospital selection”, “delay or miss in hemostatic procedure”, and “airway management” were the most commonly identified OFIs. Considering the small number of cases and uneven distribution, it was not possible to analyze the statistical differences in the frequency of OFIs by year (Table 4).

### 3.5. Transfusion Status and Transfusion Time for Patients Who Died from Hemorrhage

Among the preventable trauma deaths caused by hemorrhage, 15 patients (26.8%) did not receive blood transfusion. Among the cases where transfusion was performed, only one patient (2.4%) received transfusion within 15 min of hospital arrival. The first transfusion was found to be significantly delayed, with an average time of 3 h and 34 min and a median time of 1 h and 31 min. There was no statistical difference in the time of the first transfusion compared to that reported in the 2017 survey (Table 5).

### 3.6. Hemostatic Procedure Status and Initiation Time for Patients Who Died from Hemorrhage

Among the preventable trauma deaths caused by hemorrhage, 28 patients (50%) did not undergo procedures for hemostasis. Only one patient (3.6%) received hemostatic procedures within 1 h of arrival, with an average time of 6 h and 15 min and a median time of 3 h and 11 min, indicating a significant delay (Table 6).

### 3.7. Decompression Status and Initiation Time for Patients Who Died from Severe Brain Injury

Among the preventable cases of death from severe brain injury, decompression procedures were performed in 15 cases (40.5%). However, only three patients (20.0%) underwent the procedures within 4 h of arrival. The median time for decompression from hospital arrival was 7 h and 17 min, with an average time of 10 h and 46 min. There was no statistical difference in the rate of decompression and the time of the procedure compared to those reported in the 2017 survey (Table 7).

### 3.8. Issues in Securing a Definitive Airway for Patients Who Died from Brain Damage

Among the preventable cases of death from severe brain injury, 18 patients (48.6%) had a Glasgow Coma Scale of eight or less upon arrival. Among them, only three patients (16.7%) underwent definitive airway procedures (i.e., endotracheal intubation) within 10 min of arrival. The average time to intubation for these patients was 4 h, with a median time of 2 h and 55 min. These results were not statistically different compared to the results from 2017 (Table 8).

## 4. Discussion

In particular, there have been significant concerns regarding procedures related to hemostasis and issues associated with transfusion. Regarding transfusion, in this study, it was observed that only one case each in 2017 and 2019 complied with the “within 15 min from arrival” guideline that is recommended in the domestic regional trauma center’s standard operating procedures and evaluation criteria. Furthermore, the time to initiate the initial transfusion was found to be beyond 1 h as a significant area requiring improvement. The American College of Surgeons Committee on Trauma recommends that institutions which treat patients with severe trauma should develop their own massive transfusion protocol [19]. The introduction of a massive transfusion protocol ensures that an appropriate quantity of blood products reaches the patient within a sufficiently rapid timeframe, facilitating hemostasis by supplying the types of blood components in appropriate proportions [20,21,22,23,24,25,26]. Thus, it is crucial to focus on the development and implementation of emergency transfusion protocols targeting patients with severe trauma, and on overall quality management and improvement of the transfusion process. The term “hemostatic procedure”, highlighted by expert panels for its delay, refers to operations and interventions aimed at controlling hemorrhage. It is important to skip unnecessary tests for patients who are at risk of death due to bleeding, and immediately start hemostatic measures in the operating room or interventional radiology suite [27,28,29,30,31,32,33]; however, a significant number of patients who died from bleeding seem to have not received appropriate treatment according to these principles. This is evidenced by 29 cases in 2017 and 17 cases in 2019 being flagged as having undergone inappropriate diagnostic workup. The trauma quality improvement program of the American College of Surgeons Committee on Trauma and the evaluation criteria of domestic regional trauma centers recommend analyzing the causes of delay, when the crucial hemostatic measures exceed 1 h, and taking corrective actions. As of 2019, only one case among preventable trauma deaths due to bleeding received treatment within 1 h, with a median time of 3 h and 11 min, indicating that hospital-based trauma care quality management should be concentrated on this point. To ensure that patients with massive bleeding caused by trauma receive appropriate hemostatic measures within a sufficiently short timeframe, a skilled trauma team trained in trauma-specific interventions should be available, and facilities such as dedicated trauma resuscitation rooms, operating rooms, and interventional radiology suites need to be equipped with necessary devices. Furthermore, close collaboration with various clinical departments and teams, including readily available anesthesia teams for emergency surgery, is necessary. However, in emergency medical institutions in Korea other than regional trauma centers, where it is not feasible to maintain and utilize such resources, the “Guidelines for On-Site Emergency Treatment by 119 Rescue Workers” prioritize the transfer of patients with severe trauma to regional trauma centers based on the severity classification criteria [34]. While concentrating patients with severe trauma in regional trauma centers, efforts are also needed in nontrauma emergency medical institutions to secure facilities, equipment, and personnel resources.

According to our study, all cases of preventable trauma deaths due to central nervous system injuries were caused by traumatic brain injuries. The major OFIs identified by expert panels in this patient group were that decompressive surgery for adequate pressure relief was not promptly and appropriately performed, and proactive airway management was not adequately conducted in patients with impaired consciousness during the initial stages. The key to the treatment of traumatic brain injuries ultimately lies in successful resuscitation. For patients who are severely injured due to a trauma and developed impaired consciousness, and for whom head injuries are suspected during the initial stages, it is crucial to ensure appropriate oxygenation through immediate airway management and to maintain proper cerebral perfusion pressure through proactive hemostatic measures and resuscitation. These fundamental principles of resuscitation for patients with severe trauma can be instilled in healthcare professionals by an appropriate education, training, and effective evaluation. The Advanced Trauma Life Support course in the United States [35] has been widely reported to have improved the resuscitation procedures for patients with severe trauma, and reduced mortality rates since its implementation in healthcare professional and student education programs [36,37,38,39,40,41]. Government support is necessary for such trauma care education, and collaboration with professional societies is required to systematize and formalize sporadic education on specialized trauma resuscitation. Interpreting the results of this study and formulating and implementing health policies require consideration of several limitations. Firstly, as this is a retrospective observational study without a control group, the identified opportunities for improvements do not directly analyze the impact on specific improvement programs. Secondly, this study analyzed the treatment process of only severe trauma patients who experienced the most extreme treatment outcomes, not all trauma patients. Therefore, it may not be suitable for identifying more frequently occurring but less fatal errors. Lastly, certain aspects of the qualitative analysis relied on the consensus of expert panels rather than objective numerical data, which poses the risk of being biased based on the inclinations of the selected panel.

## 5. Conclusions

The findings of this study can be used to intuitively evaluate and address the issues that need to be focused on for improving trauma care nationwide. The major OFIs identified by expert panels indicate the need for enhancing healthcare professionals’ expertise in severe trauma treatment techniques and swiftly concentrating patients in specific specialized trauma care facilities. The OFIs highlighted in this study were identified across the prehospital, interhospital, and hospital stages, emphasizing the necessity for meaningful reviews and performance improvement activities spanning the entire patient transfer system to overcome the mentioned issues. Despite the inherent subjectivity in cost, its labor-intensive nature, and the research design, preventable trauma mortality surveillance remains a crucial tool for providing a meaningful profile of trauma care from clinical and system perspectives.

## Figures and Tables

**Figure 1 healthcare-11-02291-f001:**
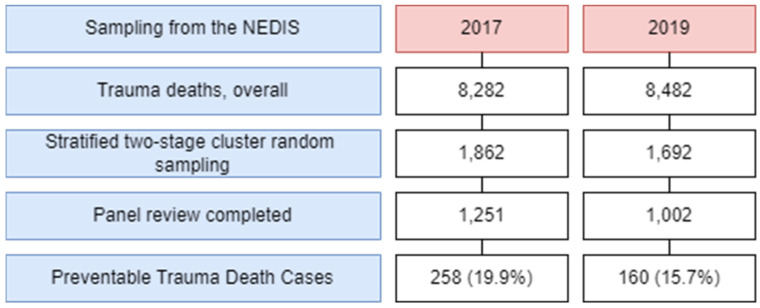
Flow chart for the comparative analysis of the preventable trauma death rates between 2017 and 2019 based on the multipanel review. NEDIS: National Emergency Department Information System.

**Table 1 healthcare-11-02291-t001:** Opportunities for improvements in preventable cases by phase of care.

Phase of Care	2017 (*n* = 258)	2019 (*n* = 160)
Prehospital	28 (10.9%)	29 (18.1%)
Interhospital transfer	58 (22.5%)	39 (24.4%)
Hospital	220 (85.3%)	144 (90.0%)
Total	306	212

Note: Some patients were noted to have multiple errors in a single phase of care.

**Table 2 healthcare-11-02291-t002:** Time from injury to admission to the destination hospital.

Analysis Group	2017			2019			*p*
	Number of Cases	Mean (SD)	Median (Quartile)	Number of Cases	Mean (SD)	Median (Quartile)	
Total cohort	1173	7 h 11 min (45 h 25 min)	45 min (26 min, 2 h 2 min)	913	9 h 55 min (63 h 17 min)	43 min (27 min, 2 h 3 min)	0.33
Transferred from another hospital	308	15 h 11 min (47 h 21 min)	3 h 2 min (1 h 54 min, 7 h 32 min)	196	32 h 52 min (125 h 41 min)	3 h 22 min (2 h 10 min, 7 h 43 min)	0.30
Directly referred to the destination hospital	865	4 h 6 min (20 h 21 min)	33 min (23 min, 55 min)	716	3 h 39 min (24 h 28 min)	35 min (25 min, 56 min)	0.65
P+PP	242	7 h 14 min (21 h 50 min)	1 h 3 min (30 min, 2 h 59 min)	144	15 h 21 min (62 h 13 min)	56 min (30 min, 3 h 46 min)	0.48
P+PP with transferred	102	14 h 56 min (38 h 57 min)	3 h 2 min (2 h 2 min, 5 h 58 min)	47	31 h 14 min (74 h 28 min)	4 h 55 min (3 h 1 min, 10 h 4 min)	0.40
P+PP with directly visited	140	1 h 37 min (5 h 1 min)	31 min (24 min, 57 min)	97	7 h 39 min (53 h 39 min)	40 min (24 min, 57 min)	0.59
P+PP with the problem in the interhospital phase	21	5 h 50 min (4 h 43 min)	2 h 49 min (2 h 2 min, 5 h 26 min)	11	38 h 9 min (95 h 43 min)	5 h 44 min (5 h 10 min, 7 h 16 min)	0.42

SD, standard deviation.

**Table 3 healthcare-11-02291-t003:** Causes of mortality in cases with preventable trauma death.

Causes	2017 Number (%)	2019 Number (%)
Hemorrhage	117 (45.3)	56 (35.0)
Sepsis/multiorgan failure	43 (16.7)	43 (26.9)
Central nerve system injury	45 (17.4)	37 (23.1)
Respiratory issues	36 (14.0)	12 (7.5)
Cardiac issues	13 (5.0)	5 (3.1)
Undetermined	4 (1.6)	7 (4.4)
Total	258 (100.0)	160 (100.0)

**Table 4 healthcare-11-02291-t004:** Opportunities for improvement by treatment stage for patients who died from hemorrhage.

Phase of Care	Inappropriate Care Related to	2017	2019
		Number (%)	Number (%)
Prehospital	Inadequate hospital selection	15 (27.8)	8 (50.0)
	Fluid resuscitation	22 (40.7)	4 (25.0)
	Delay in transfer (to first hospital)	1 (1.9)	4 (25.0)
	Delay or miss in hemostatic procedures	8 (14.8)	0 (0.0)
	Airway management	8 (14.8)	0 (0.0)
	Total	54	16
Interhospital	Delay in transfer (to final hospital)	15 (19.0)	5 (22.7)
	Delay in transfusion	16 (20.3)	4 (18.2)
	Delay in hemostatic procedures	9 (11.4)	4 (18.2)
	Unsafe transfer	5 (6.3)	3 (13.6)
	Inadequate hospital selection	2 (2.5)	2 (9.1)
	Delay in procedures other than hemostasis	17 (21.5)	2 (9.1)
	Airway management	12 (15.2)	1 (4.5)
	Inappropriate diagnostic workup	3 (3.8)	1 (4.5)
	Total	79	22
Hospital	Delay in hemostatic procedures	66 (21.5)	46 (29.3)
	Delay in procedures other than hemostasis	75 (24.4)	42 (26.8)
	Delay in Transfusion	88 (28.7)	41 (26.1)
	Inappropriate diagnostic workup	29 (9.5)	17 (10.8)
	Airway and ventilation management	25 (8.1)	6 (3.8)
	Documentation	22 (7.2)	5 (3.2)
	Others	2 (0.6)	0 (0.0)
	Total	307	157

**Table 5 healthcare-11-02291-t005:** Transfusion status and transfusion time for patients who died from hemorrhage.

Transfusion	2017 (*n* = 117)	2019 (*n* = 56)	χ^2^ (*p*)
	Number (%)	Number (%)	
Transfusion status			5.961 (0.015)
Yes	103 (88.0)	41 (73.2)	
No	14 (12.0)	15 (26.8)	
Time from visit to transfusion			0.480 (0.778)
Under 15 min	1 (1.0)	1 (2.4)	
From 15 min to 1 h	37 (36.0)	15 (36.6)	
Over 1 h	65 (63.0)	25 (61)	
Mean time from visit to transfusion (standard deviation)	3 h 37 min (±14 h 59 min)	3 h 34 min (±6 h 46 min)	
Median time from visit to transfusion (quartile)	1 h 13 min (47 min, 1 h 49 min)	1 h 31 min (47 min, 2 h 23 min)	

**Table 6 healthcare-11-02291-t006:** Hemostatic procedure status and initiation time for patients who died from hemorrhage.

Hemostatic Procedures	2017 (*n* = 117)	2019 (*n* = 56)	χ^2^ (*p*)
	Number (%)	Number (%)	
Procedure status			0.273 (0.601)
Yes	52 (44.4)	28 (50.0)	
No	65 (55.6)	28 (50.0)	
Time from visit to initiation			0.285 (0.660)
Under 1 h	5(9.6)	1 (3.6)	
Over 1 h	47 (90.4)	27 (96.4)	
Mean time from visit to initiation (standard deviation)	2 h 42 min (±1 h 24 min)	6 h 15 min (6 h 36 min)	
Median time from visit to initiation (quartile)	2 h 41 min (1 h 29 min, 3 h 22 min)	3 h 11 min (1 h 38 min, 6 h 38 min)	

**Table 7 healthcare-11-02291-t007:** Decompression status and initiation time for patients who died from severe brain injury.

Decompression	2017 (*n* = 45)	2019 (*n* = 37)	χ^2^ (*p*)
	Number (%)	Number (%)	
Procedure status			0.001 (0.978)
Yes	17 (37.8)	15 (40.5)	
No	28 (62.2)	22 (59.5)	
Time from visit to initiation			1.882 (0.576)
Under 4 h	3 (17.6)	3 (20.0)	
Over 4 h	12 (70.6)	12 (80.0)	
Unknown	2 (11.8)	0 (0.0)	
Mean time from visit to initiation (standard deviation)	48 h 47 min (±93 h 28 min)	10 h 46 min (±7 h 10 min)	
Median time from visit to initiation (quartile)	9 h 48 min (5 h 46 min, 21 h 28 min)	7 h 17 min (5 h 31 min, 16 h 10 min)	

**Table 8 healthcare-11-02291-t008:** Initial mental status and intubation time for patients who died from severe brain injury.

	2017 (*n* = 45)	2019 (*n* = 37)	χ^2^ (*p*)
	Number (%)	Number (%)	
Decreased mentality ^a^			
Yes	21 (46.7)	18 (48.6)	
No	24 (53.3)	19 (51.4)	
Time from visit to intubation			5.780 (0.124)
Under 10 min	5 (23.9)	3 (16.7)	
Over 10 min under 1 h	10 (47.6)	4 (22.2)	
Over 1 h	4 (19.0)	10 (55.6)	
Unknown	2 (9.5)	1 (5.6)	
Mean time from visit to intubation (standard deviation)	48 h 47 min (±93 h 28 min)	4 h (±4 h 35 min)	
Median time from visit to intubation (quartile)	9 h 48 min (5 h 46 min, 21 h 28 min)	2 h 55 min (45 min, 5 h 32 min)	

^a^ Glasgow Coma Scale ≤ 8, P or U in APVU system (alert, painful stimuli, verbal stimuli, unresponsive).

## Data Availability

The data used in this study contain personal sensitive information and cannot be publicly disclosed. However, they can be made available upon request to the corresponding author.

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
