# Peer review of "National Assessment of Opportunities for Improvement in Preventable Trauma Deaths: A Mixed-Methods Study"

_healthcare, 2023, doi:10.3390/healthcare11162291_

Round 1
Reviewer 1 Report
The manuscript entitled “National Assessment of Opportunities for Improvement in Preventable Trauma Deaths: A Qualitative Investigation” was interesting. The following comments can help the authors to improve it:
1- In the Abstract, It is better to start with some general statements as an introduction, then continue it with the study design, research participants, research instrument, data collection and analysis.
2- In the methods section, it is better to apply the above suggested sequence for subheadings and the content. Moreover, if the study composed of different phases, please name them and clarify each phase with all related details.
3- I assume it is better to say “a mixed-methods study” rather than “a qualitative investigation” in the title, as both qualitative and quantitative approaches were used.
4- In Figure 1, please add a percentage for the final preventable death in 2017.
5- The discussion section needs to be supported by more related studies.
6- Please add the research limitations.
7- Some references are older than 10 years ago, please replace them with the more recent ones.
Author Response
Point 1: In the Abstract, It is better to start with some general statements as an introduction, then continue it with the study design, research participants, research instrument, data collection and analysis.
Response 1: Thank you for the comment regarding the structure of the Abstract. According to the comment, we have revised the abstract as follows:
(General statements) Trauma is a significant public health issue worldwide, particularly affecting economically active age groups. Quality management of trauma care at the national level is crucial to improve outcomes of major trauma. (Study design) In Korea, a biennial nationwide survey on preventable trauma death rate is conducted. Based on the survey results, we analyzed opportunities for improving the trauma treatment process. (Research participants and Data collection) Expert panels reviewed records of 8,282 and 8,482 trauma-related deaths in 2017 and 2019, respectively, identifying 258 and 160 cases in each year as preventable deaths. (Research instrument and Analysis) Opportunities for improvement were categorized into prehospital, inter-hospital, and hospital stages. Hemorrhage was the primary cause of death, followed by sepsis/multi-organ failure and central nervous system injury. Delayed hemostatic procedures and transfusions were common areas for improvement in hospital stage. Inter-hospital transfers experienced significant delays in arrival time. This study emphasizes the need to enhance trauma care by refining treatment techniques, centralizing patients in specialized facilities, and implementing comprehensive reviews and performance improvements throughout the patient transfer system. The findings offer valuable insights for addressing trauma care improvement from both clinical and systemic perspectives.
Point 2: In the methods section, it is better to apply the above suggested sequence for subheadings and the content. Moreover, if the study composed of different phases, please name them and clarify each phase with all related details.
Response 2: Based on the reviewer's suggestions, we have restructured the method section. Utilizing the recommended subheadings provided by the reviewer, we made slight modifications to the content, aiming to provide detailed explanations of the research methodology. The following is the newly reorganized method section.
Study design - Population and Data Sampling
This study utilized data from patients who visited hospitals and subsequently died due to trauma in 2017 and 2019. Patients who experienced trauma were defined as those with one or more S or T codes according to the Korean Standard Classification of Diseases, seventh edition [17]. The national emergency department information system (NEDIS) was used as a sampling framework for retrieving mortality statistics for patients who experienced trauma [18]. NEDIS is a national database that includes clinical and administrative data of all patients visiting emergency departments. In 2017 and 2019, there were 8,282 and 8,482 trauma-related deaths, respectively, as aggregated from emergency medical institutions nationwide. To obtain an unbiased sample that reflects the characteristics of the population, a stratified two-stage cluster sampling method with stratification and dual-stage clustering was employed. For the first stage of stratification, variables such as region, hospital type (regional trauma center, regional emergency medical center, and local emergency medical institution), and the number of deaths (≥ 100, 50–99, 30–49, 10–29, and <10) were used. For the second stage of stratification, variables including time of death and patient age were used (Figure 1).
Research participants - Data Collection Method and Types
Based on the stratified two-stage cluster random sampling, the research team re-quested medical records of the selected patients from the hospitals where they were treat-ed; records of a sample of 1,862 and 1,692 patients were retrieved for the 2017 and 2019 cohorts, respectively. The requested medical records were submitted to the Ministry of Health and Welfare through the hospitals in accordance with the Emergency Medical Services Act and the Medical Act. Important imaging examination results were stored on compact disks and submitted with their interpretations. The requested medical records included initial emergency room visit records, progress notes, nursing charts, official imaging interpretation sheets, blood test results, and discharge records. Additionally, ambulance records were collected for the analysis of prehospital stage OFIs.
Research instrument - Multidisciplinary Case Review and Preventable Trauma Death Rate
To review medical records, the research team developed a structured review form based on the data sheet proposed by the World Health Organization guidelines [4] (Supplement 1). A total of 32 preliminary surveyors, mainly trauma coordinators working at regional trauma centers, organized and summarized the medical records before the expert panel review. All surveyors had completed training on case reviews provided by the National Medical Center. To identify OFIs in reviewed cases and assess preventability, a panel of trauma care specialists consisting of 25 trauma physicians working mainly at regional trauma centers was selected. After independently reviewing each case, they participated in discussions in groups composed of two general surgeons, one thoracic surgeon, one neurosurgeon, and one emergency physician. The panel members assessed the preventability of trauma deaths for 1,251 cases in 2017 and 1,208 cases in 2019, excluding cases that were determined not to be trauma-related deaths. The final calculated PTDR was 19.9% in 2017 and 15.7% in 2019. It should be noted that the PTDR was calculated using weighted estimates, rather than actual numbers, and may differ from calculations using actual numbers.
Analysis - Qualitative Analysis of Preventable Trauma Death Cases
After determining the preventability for each patient with a trauma-related death, the panel identified OFIs in the patient's treatment process. The identified OFIs were categorized into prehospital, inter-hospital, and hospital stages. The preventable trauma death cases were classified according to the primary causes of death, and the results were com-pared with those of a previous study conducted on deceased individuals in 2017 [8]. The performance and timing of major procedures such as transfusion, hemostasis surgeries, craniectomy, and intubation, which were performed for patients who died from bleeding or severe brain injury, were examined.
Ethics Statement
The study was approved by the Ajou University Institutional Review Board, and informed consent was waived due to the observational nature of the study.
Point 3: I assume it is better to say “a mixed-methods study” rather than “a qualitative investigation” in the title, as both qualitative and quantitative approaches were used.
Response 3: Following the reviewer's advice, we have revised the title as follows.
Before: National Assessment of Opportunities for Improvement in Preventable Trauma Deaths: A Qualitative Investigation
After: National Assessment of Opportunities for Improvement in Preventable Trauma Deaths: A Mixed-Methods study
Point 4: In Figure 1, please add a percentage for the final preventable death in 2017.
Response 4: As per the reviewer's suggestion, we have added the rate to the 'preventable trauma death cases' cell of Figure 1 for the year 2017. As documented in the 'Materials and Methods' section, the Preventable trauma death rate was calculated using 'weighted estimates,' which means the results may not align with the calculations based on the actual number of cases. For instance, in 2019, without applying the weighting, the actual number of preventable trauma death cases was 160 in total, but after applying the weighting, it increased to 190 cases. Comparing this to the estimated population with a total of 1208 cases, the percentage derived is 15.7%.
Before:
After:
Point 5: The discussion section needs to be supported by more related studies.
Point 7: Some references are older than 10 years ago, please replace them with the more recent ones.
Response 5 and 7:
We replaced all the papers before 2012 with recently published papers from the last 10 years, excluding essential papers related to the theoretical background of the Preventable trauma death rate, WHO and Korean statistics-related literature. Accordingly, the previous papers 22,23,24,25 and 27 were removed, and papers 20-35, and 37-41 were added.
The following are the references that have been removed from the original document. (Citation numbers correspond to the pre-revised version.)
- Ariyanayagam, D.C.; Naraynsingh, V.; Maraj, I. The impact of the ATLS course on traffic accident mortality in Trinidad and Tobago. West Indian Med J. 1992, 41, 72–74.
- Ali, J.; Adam, R.; Butler, A.K.; Chang, H.; Howard, M.; Gonsalves, D.; Pitt-Miller, P.; Stedman, M.; Winn, J.; Williams, J.I. Trauma outcome improves following the advanced trauma life support program in a developing country. J Trauma. 1993, 34, 890–8; discussion 898. DOI:10.1097/00005373-199306000-00022.
- van Olden, G.D.; Meeuwis, J.D.; Bolhuis, H.W.; Boxma, H.; Goris, R.J. Advanced trauma life support study: quality of diagnostic and therapeutic procedures. J Trauma. 2004, 57, 381–384. DOI:10.1097/01.ta.0000096645.13484.e6.
- van Olden, G.D.; Meeuwis, J.D.; Bolhuis, H.W.; Boxma, H.; Goris, R.J. Clinical impact of advanced trauma life support. Am J Emerg Med. 2004, 22, 522–525. DOI:10.1016/j.ajem.2004.08.013.
- Hedges, J.R.; Adams, A.L.; Gunnels, M.D. ATLS practices and survival at rural level III trauma hospitals, 1995–1999. Prehosp Emerg Care. 2002, 6, 299–305. DOI:10.1080/10903120290938337.
The following are the newly added references. (Citation numbers correspond to the revised version.)
- Kwon, J.; Yoo, J.; Kim, S.; Jung, K.; Yi, I.K. Evaluation of the Potential for Improvement of Clinical Outcomes in Trauma Patients with Massive Hemorrhage by Maintaining a High Plasma-to-Red Blood Cell Ratio during the First Hour of Hospitalization. Emerg Med Int. 2023, Published online. DOI:10.1155/2023/5588707
- Holcomb, J.B.; del Junco, D.J.; Fox, E.E.; Wade, C.E.; Cohen, M.J.; Schreiber, M.A.; Alarcon, L.H.; Bai, Y.; Brasel, K.J.; Bulger, E.M.; et al. The prospective, observational, multicenter, major trauma transfusion (PROMMTT) study: comparative effectiveness of a time-varying treatment with competing risks. JAMA Surg. 2013, 148(2):127–136. DOI:10.1001/2013.jamasurg.387.
- Holcomb, J.B.; Tilley, B.C.; Baraniuk, S.; Fox, E.E.; Wade, C.E.; Podbielski, J.M.; del Junco, D.J.; Brasel, K.J.; Bulger, E.M.; Callcut, R.A.; et al. Transfusion of plasma, platelets, and red blood cells in a 1:1:1 vs a 1:1:2 ratio and mortality in patients with severe trauma: the PROPPR randomized clinical trial. JAMA. 2015, 313(5):471–482. DOI:10.1001/jama.2015.12.
- Neff, L.P.; Cannon, J.W.; Morrison, J.J.; Edwards, M.J.; Spinella, P.C.; Borgman, M.A. Clearly defining pediatric massive transfusion: cutting through the fog and friction with combat data. J Trauma Acute Care Surg. 2015, 78(1):22–28; discussion 28–29. DOI:10.1097/TA.0000000000000488.
- Khan, S.; Allard, S.; Weaver, A.; Barber, C.; Davenport, R.; Brohi, K. A major haemorrhage protocol improves the delivery of blood component therapy and reduces waste in trauma massive transfusion. Injury. 2013, 44(5):587–592. DOI: 10.1016/j.injury.2012.09.029.
- Kutcher, M.E.; Kornblith, L.Z.; Narayan, R.; Curd, V.; Daley, A.T.; Redick, B.J.; Nelson, M.F.; Fiebig, E.W.; Cohen, M.J. A paradigm shift in trauma resuscitation: evaluation of evolving massive transfusion practices. JAMA Surg. 2013, 148(9):834–840. DOI:10.1001/jamasurg.2013.2911.
- Bawazeer, M.; Ahmed, N.; Izadi, H.; McFarlan, A.; Nathens, A.; Pavenski, K. Compliance with a massive transfusion protocol (MTP) impacts patient outcome. Injury. 2015, 46(1):21–28. DOI:10.1016/j.injury.2014.09.020.
- Latif, R.K.; Clifford, S.P.; Baker, J.A.; Lenhardt, R.; Haq, M.Z.; Huang, J.; Farah, I.; Businger, J.R. Traumatic hemorrhage and chain of survival. Scand J Trauma Resusc Emerg Med. 2023, 24;31(1):25. DOI:10.1186/s13049-023-01088-8.
- Loftus, T.J.; Efron, P.A.; Bala, T.M.; Rosenthal, M.D.; Croft, C.A.; Walters, M.S.; Smith, R.S.; Moore, F.A.; Mohr, A.M.; Brakenridge, S.C. The impact of standardized protocol implementation for surgical damage control and temporary abdominal closure after emergent laparotomy. J Trauma Acute Care Surg. 2019, 86(4):670-8. DOI:10.1097/TA.0000000000002170.
- Smith, J.W.; Matheson, P.J.; Franklin, G.A.; Harbrecht, B.G.; Richardson, J.D.; Garrison, R.N. Randomized Controlled Trial Evaluating the Efficacy of Peritoneal Resuscitation in the Management of Trauma Patients Undergoing Damage Control Surgery. J Am Coll Surg. 2017, 224(4):396-404. DOI:10.1016/j.jamcollsurg.2016.12.047.
- Roberts, D.J.; Bobrovitz, N.; Zygun, D.A.; Ball, C.G.; Kirkpatrick, A.W.; Faris, P.D.; Brohi, K.; D’Amours, S.; Fabian, T.C.; Inaba, K.; et al. Indications for Use of Damage Control Surgery in Civilian Trauma Patients: A Content Analysis and Expert Appropriateness Rating Study. Ann Surg. 2016, 263(5):1018-27. DOI:10.1097/SLA.0000000000001347.
- Alarhayem, A.Q.; Myers, J.G.; Dent, D.; Liao, L.; Muir, M.; Mueller, D.; Nicholson, S.; Cestero, R.; Johnson, M.C.; Stewart, R.; et al. Time is the enemy: mortality in trauma patients with hemorrhage from torso injury occurs long before the "golden hour". Am J Surg . 2016, 212(6):1101–1105. DOI:10.1016/j.amjsurg.2016.08.018
- Adams, D.; McDonald, P.L.; Sullo, E.; Merkle, A.B.; Nunez, T.; Sarani, B.; Shackelford, S.A.; Bowyer, M.W.; van der Wees, P. Management of non-compressible torso hemorrhage of the abdomen in civilian and military austere/remote environments: protocol for a scoping review. Trauma Surg Acute Care Open. 2021, 19, 6(1):e000811. DOI:10.1136/tsaco-2021-000811.
- Allen, S.R.; Scantling, D.R.; Delgado, M.K.; Mancini, J.; Holena, D.N.; Kim, P.; Pascual, J.L.; Reilly, P. Penetrating torso injuries in older adults: increased mortality likely due to "failure to rescue". Eur J Trauma Emerg Surg. 2015, 41(6):657-63. DOI:10.1007/s00068-014-0491-7.
- National Emergency Management Agency. The Standard Protocols for 119 Emergency Medical Services Providers; National Emergency Management Agency, 2019.
- American College of Surgeons. Advanced Trauma Life Support (ATLS), 10th ed.; American College of Surgeons Committee on Trauma, 2018.
- Hashmi, Z.G.; Haider, A.H.; Zafar, S.N.; Kisat, M.; Moosa, A.; Siddiqui, F.; Pardhan, A.; Latif, A.; Zafar, H. Hospital-based trauma quality improvement initiatives: first step toward improving trauma outcomes in the developing world. J Trauma Acute Care Surg. 2013, 75, 60–8; discussion 68. DOI:10.1097/TA.0b013e31829880a0.
- Mitchnik, I.Y.; Regev, S.; Rivkind, A.I.; Fogel, I. Disparities in trauma care education: An observational study of the ATLS course within a national trauma system. Injury. 2023, 7;110860. DOI:10.1016/j.injury.2023.110860.
- Neutel, E.; Kuhn, S.; Driscoll, P.; Gwinnutt, C.; Moreira, Z.; Veloso, A.; Manso, M.C.; Carneiro, A. Does participation in the European Trauma Course lead to new behaviours and organisational change? A Portuguese experience. BMC Med Educ. 2023, 6, 23(1), 415. DOI:10.1186/s12909-023-04322-0.
- Archuleta, M.; McGraw, C.; D'Huyvetter, C.; Mains, C.W. An Educational Outreach Program: A Trauma System's 5-Year Experience. J Trauma Nurs. 2022, 29(3), 152-157. DOI:10.1097/JTN.0000000000000653.
- Wärnberg, M.G.; Berg, J.; Bhandarkar, P.; Chatterjee, A.; Chatterjee, S.; Chintamani, C.; Felländer-Tsai, L.; Gadgil, A.; Ghag, G.; Hasselberg, M.; et al. A pilot multicentre cluster randomised trial to compare the effect of trauma life support training programmes on patient and provider outcomes. BMJ Open. 2022, 18, 12(4):e057504. DOI:10.1136/bmjopen-2021-057504.
- van Maarseveen, O.E.C.; Ham, W.H.W.; van de Ven, N.L.M.; Saris, T.F.F.; Leenen, L.P.H. Effects of the application of a checklist during trauma resuscitations on ATLS adherence, team performance, and patient-related outcomes: a systematic review. Eur J Trauma Emerg Surg. 2020, 46(1), 65-72. DOI:10.1007/s00068-019-01181-7.
In addition to this, we have added the following sentences to lines 220 to 223 in the discussion section.
The introduction of a massive transfusion protocol ensures that an appropriate quantity of blood products reaches the patient within a sufficiently rapid timeframe, facilitating hemostasis by supplying the types of blood components in appropriate proportions.
Point 6: Please add the research limitations.
Response 6: Please provide your response for Point 2. (in red)
As per the reviewer's suggestion, we have added the following limitations to the last part of the ‘Discussion’ section.
Interpreting the results of this study and formulating and implementing health policies require consideration of several limitations. Firstly, as this is a retrospective observational study without a control group, the identified opportunities for improvements do not directly analyze the impact on specific improvement programs. Secondly, this study analyzed the treatment process of only severe trauma patients who experienced the most extreme treatment outcomes, not all trauma patients. Therefore, it may not be suitable for identifying more frequently occurring but less fatal errors. Lastly, certain aspects of the qualitative analysis relied on the consensus of expert panels rather than objective numerical data, which poses the risk of being biased based on the inclinations of the selected panel.

Reviewer 2 Report
This is an interesting and valualbe study, here, this study analyzed opportunities for improvement in trauma care, and the author found that this preventable trauma mortality surveillance study emphasizes the need to enhance trauma care by improving treatment techniques, concentrating patients in specialized facilities, and implementing comprehensive reviews and performance improvement activities across the entire patient transfer system. So, this study provide valuable insights for addressing areas that need trauma care improvement from both clinical and system perspectives.
Why just show the data in 2017 and 2019 ?What about these data in 2018 ?
Please also improve the quality of Discussion part, including refs 19-21.
Author Response
Point 1: This is an interesting and valualbe study, here, this study analyzed opportunities for improvement in trauma care, and the author found that this preventable trauma mortality surveillance study emphasizes the need to enhance trauma care by improving treatment techniques, concentrating patients in specialized facilities, and implementing comprehensive reviews and performance improvement activities across the entire patient transfer system. So, this study provide valuable insights for addressing areas that need trauma care improvement from both clinical and system perspectives.
Why just show the data in 2017 and 2019 ?What about these data in 2018 ?
Response 1: Thank you for your favorable assessment of our study's findings. This research was conducted based on the results of the "Nation-Wide Preventable Trauma Death Survey," which is carried out every two years by the Ministry of Health and Welfare in Korea to assess the national performance of trauma care. This national survey was initially planned as an intermediate evaluation for the long-term task of establishing a trauma care system and is conducted once every odd-numbered year. Therefore, there are no results available for the year 2018. Nevertheless, despite this, the research team empathizes with the reviewer's suggestions and is currently planning to adjust the sample size and simplify the methods to conduct the survey more frequently.
Point 2: Please also improve the quality of Discussion part, including refs 19-21.
Response 2: As pointed out by the reviewer, this discussion contains some leaps in reasoning in certain parts, and some references do not sufficiently support the arguments or are outdated for over 10 years. Therefore, the authors have added sentences to the discussion and replaced the references accordingly.
The following are the newly inserted sentences in lines 220 to 223 of the discussion section.
The introduction of a massive transfusion protocol ensures that an appropriate quantity of blood products reaches the patient within a sufficiently rapid timeframe, facilitating hemostasis by supplying the types of blood components in appropriate proportions [20-26].
we have added the following limitations to the last part of the ‘Discussion’ section.
Interpreting the results of this study and formulating and implementing health policies require consideration of several limitations. Firstly, as this is a retrospective observational study without a control group, the identified opportunities for improvements do not directly analyze the impact on specific improvement programs. Secondly, this study analyzed the treatment process of only severe trauma patients who experienced the most extreme treatment outcomes, not all trauma patients. Therefore, it may not be suitable for identifying more frequently occurring but less fatal errors. Lastly, certain aspects of the qualitative analysis relied on the consensus of expert panels rather than objective numerical data, which poses the risk of being biased based on the inclinations of the selected panel.
We replaced all the papers before 2012 with recently published papers from the last 10 years, excluding essential papers related to the theoretical background of the Preventable trauma death rate, WHO and Korean statistics-related literature. Accordingly, the previous papers 22,23,24,25 and 27 were removed, and papers 20-35, and 37-41 were added.
The following are the references that have been removed from the original document. (Citation numbers correspond to the pre-revised version.)
- Ariyanayagam, D.C.; Naraynsingh, V.; Maraj, I. The impact of the ATLS course on traffic accident mortality in Trinidad and Tobago. West Indian Med J. 1992, 41, 72–74.
- Ali, J.; Adam, R.; Butler, A.K.; Chang, H.; Howard, M.; Gonsalves, D.; Pitt-Miller, P.; Stedman, M.; Winn, J.; Williams, J.I. Trauma outcome improves following the advanced trauma life support program in a developing country. J Trauma. 1993, 34, 890–8; discussion 898. DOI:10.1097/00005373-199306000-00022.
- van Olden, G.D.; Meeuwis, J.D.; Bolhuis, H.W.; Boxma, H.; Goris, R.J. Advanced trauma life support study: quality of diagnostic and therapeutic procedures. J Trauma. 2004, 57, 381–384. DOI:10.1097/01.ta.0000096645.13484.e6.
- van Olden, G.D.; Meeuwis, J.D.; Bolhuis, H.W.; Boxma, H.; Goris, R.J. Clinical impact of advanced trauma life support. Am J Emerg Med. 2004, 22, 522–525. DOI:10.1016/j.ajem.2004.08.013.
- Hedges, J.R.; Adams, A.L.; Gunnels, M.D. ATLS practices and survival at rural level III trauma hospitals, 1995–1999. Prehosp Emerg Care. 2002, 6, 299–305. DOI:10.1080/10903120290938337.
The following are the newly added references. (Citation numbers correspond to the revised version.)
- Kwon, J.; Yoo, J.; Kim, S.; Jung, K.; Yi, I.K. Evaluation of the Potential for Improvement of Clinical Outcomes in Trauma Patients with Massive Hemorrhage by Maintaining a High Plasma-to-Red Blood Cell Ratio during the First Hour of Hospitalization. Emerg Med Int. 2023, Published online. DOI:10.1155/2023/5588707
- Holcomb, J.B.; del Junco, D.J.; Fox, E.E.; Wade, C.E.; Cohen, M.J.; Schreiber, M.A.; Alarcon, L.H.; Bai, Y.; Brasel, K.J.; Bulger, E.M.; et al. The prospective, observational, multicenter, major trauma transfusion (PROMMTT) study: comparative effectiveness of a time-varying treatment with competing risks. JAMA Surg. 2013, 148(2):127–136. DOI:10.1001/2013.jamasurg.387.
- Holcomb, J.B.; Tilley, B.C.; Baraniuk, S.; Fox, E.E.; Wade, C.E.; Podbielski, J.M.; del Junco, D.J.; Brasel, K.J.; Bulger, E.M.; Callcut, R.A.; et al. Transfusion of plasma, platelets, and red blood cells in a 1:1:1 vs a 1:1:2 ratio and mortality in patients with severe trauma: the PROPPR randomized clinical trial. JAMA. 2015, 313(5):471–482. DOI:10.1001/jama.2015.12.
- Neff, L.P.; Cannon, J.W.; Morrison, J.J.; Edwards, M.J.; Spinella, P.C.; Borgman, M.A. Clearly defining pediatric massive transfusion: cutting through the fog and friction with combat data. J Trauma Acute Care Surg. 2015, 78(1):22–28; discussion 28–29. DOI:10.1097/TA.0000000000000488.
- Khan, S.; Allard, S.; Weaver, A.; Barber, C.; Davenport, R.; Brohi, K. A major haemorrhage protocol improves the delivery of blood component therapy and reduces waste in trauma massive transfusion. Injury. 2013, 44(5):587–592. DOI: 10.1016/j.injury.2012.09.029.
- Kutcher, M.E.; Kornblith, L.Z.; Narayan, R.; Curd, V.; Daley, A.T.; Redick, B.J.; Nelson, M.F.; Fiebig, E.W.; Cohen, M.J. A paradigm shift in trauma resuscitation: evaluation of evolving massive transfusion practices. JAMA Surg. 2013, 148(9):834–840. DOI:10.1001/jamasurg.2013.2911.
- Bawazeer, M.; Ahmed, N.; Izadi, H.; McFarlan, A.; Nathens, A.; Pavenski, K. Compliance with a massive transfusion protocol (MTP) impacts patient outcome. Injury. 2015, 46(1):21–28. DOI:10.1016/j.injury.2014.09.020.
- Latif, R.K.; Clifford, S.P.; Baker, J.A.; Lenhardt, R.; Haq, M.Z.; Huang, J.; Farah, I.; Businger, J.R. Traumatic hemorrhage and chain of survival. Scand J Trauma Resusc Emerg Med. 2023, 24;31(1):25. DOI:10.1186/s13049-023-01088-8.
- Loftus, T.J.; Efron, P.A.; Bala, T.M.; Rosenthal, M.D.; Croft, C.A.; Walters, M.S.; Smith, R.S.; Moore, F.A.; Mohr, A.M.; Brakenridge, S.C. The impact of standardized protocol implementation for surgical damage control and temporary abdominal closure after emergent laparotomy. J Trauma Acute Care Surg. 2019, 86(4):670-8. DOI:10.1097/TA.0000000000002170.
- Smith, J.W.; Matheson, P.J.; Franklin, G.A.; Harbrecht, B.G.; Richardson, J.D.; Garrison, R.N. Randomized Controlled Trial Evaluating the Efficacy of Peritoneal Resuscitation in the Management of Trauma Patients Undergoing Damage Control Surgery. J Am Coll Surg. 2017, 224(4):396-404. DOI:10.1016/j.jamcollsurg.2016.12.047.
- Roberts, D.J.; Bobrovitz, N.; Zygun, D.A.; Ball, C.G.; Kirkpatrick, A.W.; Faris, P.D.; Brohi, K.; D’Amours, S.; Fabian, T.C.; Inaba, K.; et al. Indications for Use of Damage Control Surgery in Civilian Trauma Patients: A Content Analysis and Expert Appropriateness Rating Study. Ann Surg. 2016, 263(5):1018-27. DOI:10.1097/SLA.0000000000001347.
- Alarhayem, A.Q.; Myers, J.G.; Dent, D.; Liao, L.; Muir, M.; Mueller, D.; Nicholson, S.; Cestero, R.; Johnson, M.C.; Stewart, R.; et al. Time is the enemy: mortality in trauma patients with hemorrhage from torso injury occurs long before the "golden hour". Am J Surg . 2016, 212(6):1101–1105. DOI:10.1016/j.amjsurg.2016.08.018
- Adams, D.; McDonald, P.L.; Sullo, E.; Merkle, A.B.; Nunez, T.; Sarani, B.; Shackelford, S.A.; Bowyer, M.W.; van der Wees, P. Management of non-compressible torso hemorrhage of the abdomen in civilian and military austere/remote environments: protocol for a scoping review. Trauma Surg Acute Care Open. 2021, 19, 6(1):e000811. DOI:10.1136/tsaco-2021-000811.
- Allen, S.R.; Scantling, D.R.; Delgado, M.K.; Mancini, J.; Holena, D.N.; Kim, P.; Pascual, J.L.; Reilly, P. Penetrating torso injuries in older adults: increased mortality likely due to "failure to rescue". Eur J Trauma Emerg Surg. 2015, 41(6):657-63. DOI:10.1007/s00068-014-0491-7.
- National Emergency Management Agency. The Standard Protocols for 119 Emergency Medical Services Providers; National Emergency Management Agency, 2019.
- American College of Surgeons. Advanced Trauma Life Support (ATLS), 10th ed.; American College of Surgeons Committee on Trauma, 2018.
- Hashmi, Z.G.; Haider, A.H.; Zafar, S.N.; Kisat, M.; Moosa, A.; Siddiqui, F.; Pardhan, A.; Latif, A.; Zafar, H. Hospital-based trauma quality improvement initiatives: first step toward improving trauma outcomes in the developing world. J Trauma Acute Care Surg. 2013, 75, 60–8; discussion 68. DOI:10.1097/TA.0b013e31829880a0.
- Mitchnik, I.Y.; Regev, S.; Rivkind, A.I.; Fogel, I. Disparities in trauma care education: An observational study of the ATLS course within a national trauma system. Injury. 2023, 7;110860. DOI:10.1016/j.injury.2023.110860.
- Neutel, E.; Kuhn, S.; Driscoll, P.; Gwinnutt, C.; Moreira, Z.; Veloso, A.; Manso, M.C.; Carneiro, A. Does participation in the European Trauma Course lead to new behaviours and organisational change? A Portuguese experience. BMC Med Educ. 2023, 6, 23(1), 415. DOI:10.1186/s12909-023-04322-0.
- Archuleta, M.; McGraw, C.; D'Huyvetter, C.; Mains, C.W. An Educational Outreach Program: A Trauma System's 5-Year Experience. J Trauma Nurs. 2022, 29(3), 152-157. DOI:10.1097/JTN.0000000000000653.
- Wärnberg, M.G.; Berg, J.; Bhandarkar, P.; Chatterjee, A.; Chatterjee, S.; Chintamani, C.; Felländer-Tsai, L.; Gadgil, A.; Ghag, G.; Hasselberg, M.; et al. A pilot multicentre cluster randomised trial to compare the effect of trauma life support training programmes on patient and provider outcomes. BMJ Open. 2022, 18, 12(4):e057504. DOI:10.1136/bmjopen-2021-057504.
- van Maarseveen, O.E.C.; Ham, W.H.W.; van de Ven, N.L.M.; Saris, T.F.F.; Leenen, L.P.H. Effects of the application of a checklist during trauma resuscitations on ATLS adherence, team performance, and patient-related outcomes: a systematic review. Eur J Trauma Emerg Surg. 2020, 46(1), 65-72. DOI:10.1007/s00068-019-01181-7.

Reviewer 3 Report
Dear Authors,
I appreciate the opportunity to review Your work. I congratulate a very good study. It shows what needs to be corrected in the health system to improve outcomes in trauma patients. The article is professionally written, consistent.
Two minor concerns:
Firstly, how was the stratified two-stage cluster random sampling done? No randomizer nor statistical program was mentioned.
Secondly, why specifically 2017 and 2019 were chosen?
Regards,
Reviewer
Author Response
Point 1: Firstly, how was the stratified two-stage cluster random sampling done? No randomizer nor statistical program was mentioned.
Response 1: Thank you for your favorable assessment of our study's findings. To calculate a representative nationwide Preventable Trauma Death Rate, a statistical stratification process was used to sample the study subjects. The first stratification was based on medical institutions, while the second stratification was related to characteristics of the deceased. The first-stage stratification included variables such as the region of the medical institution, type of emergency medical institution, and number of trauma-related deaths per medical institution. The second-stage stratification included variables such as time of death and age of the deceased.
Stratification Categories:
- First-stage stratification (Medical Institution):
- Region (7): Seoul, Gyeonggi/Incheon, Daejeon/Chungcheong, Gangwon, Honam, Yeongnam, Jeju
- Emergency Medical Institution Type (3): Regional Emergency Medical Center/Regional Trauma Center, Local Emergency Medical Center, Local Emergency Medical Institution
- Number of Deaths (2): Less than 30 trauma deaths, 30 or more trauma deaths
- Second-stage stratification (Deceased):
- Time of Death (4): Before hospital arrival, In the emergency room, After admission, After transfer
- Age (3): Under 15 years, 15-54 years, 55 years and older
The target sample size for this study was set at 1,300 individuals. It was estimated that a sample size of approximately 1,300 individuals would ensure stable estimates with an error margin of about ±3.3 percentage points at a 95% confidence level for the estimated preventable trauma death rate. (In the 2015 survey, the completed sample size was 943 individuals, with an error margin of ±4.5 percentage points at a 95% confidence level, while in the 2017 survey, the completed sample size was 1,232 individuals, with an error margin of ±3.8 percentage points at a 95% confidence level.)
The distribution of samples according to the type of emergency medical institution was based on proportional allocation, but the final sample size within each emergency medical institution was determined according to the number of trauma deaths.
The extraction of sample institutions used a stratified random sampling method based on factors such as region, type of emergency medical institution, and number of trauma deaths. Within the sample institutions, the extraction of trauma-related deaths for the study used a stratified random sampling method based on factors such as age group and time of death.
Due to space constraints, the above information was summarized in a supplementary table as it was not feasible to directly insert it into the paper.
Supplementary Table. Summary of Trauma Patient Sample Design and Estimation Method
|
Item |
Content |
|
|
Population |
Target Population |
Trauma patients who died after visiting domestic emergency medical institutions |
|
Survey Population |
Deaths within one year in emergency medical institutions with at least one death |
|
|
Sampling Unit |
1st Stage Sampling Unit (PSU) |
Emergency medical institution |
|
2nd Stage Sampling Unit (SSU) |
Deceased patients within emergency medical institutions |
|
|
Stratification |
1st Stage Stratification Variables (Medical Institution Stratification) |
â–ª Region (7): Seoul, Gyeonggi/Incheon, Daejeon/Chungcheong, Gangwon, Honam, Yeongnam, Jeju â–ª Emergency medical institution level (3): Regional emergency medical center/regional trauma center, Local emergency medical center, Local emergency medical institution â–ª Number of deaths (2): Less than 30 trauma deaths, 30 or more trauma deaths |
|
2nd Stage Stratification Variables (Deceased Patient Stratification) |
â–ª Time of death (4): Death before arriving at the hospital, Death in the emergency room, Death after admission, Death after transfer â–ª Patient age (3): Under 15 years, 15-54 years, 55 years and older |
|
|
Sample Size |
â–ª The total sample size was determined to be approximately 1,700 individuals after considering the target error level. â–ª It is expected to provide stable estimates with a margin of error of approximately 3.3% at a 95% confidence level for estimating the preventable trauma death rate (mobility rate). |
|
|
Sample Extraction |
â–ª A stratified random sampling of 279 sample medical institutions was conducted. â–ª Deceased trauma patients were randomly sampled within each sample hospital, resulting in a total of 1,692 individuals. (1,862 in 2017) |
|
|
Weight Calculation |
Weight was calculated for each hospital and deceased patient based on the sampling design. |
|
|
Estimation |
â–ª The estimated preventable trauma death rate was calculated using the weighted estimate based on the weight. â–ª The standard error and margin of error for the estimated preventable trauma death rate were calculated. |
|
Point 2: Secondly, why specifically 2017 and 2019 were chosen?
Response 2: This research was conducted based on the results of the "Nation-Wide Preventable Trauma Death Survey," which is carried out every two years by the Ministry of Health and Welfare in Korea to assess the national performance of trauma care. This national survey was initially planned as an intermediate evaluation for the long-term task of establishing a trauma care system and is conducted once every odd-numbered year. Therefore, there are no results available for the year 2018. Nevertheless, despite this, the research team empathizes with the reviewer's suggestions and is currently planning to adjust the sample size and simplify the methods to conduct the survey more frequently. As a note, the research team is currently conducting a study focusing on deceased patients in 2021, and plans to present the research findings later this year.
Reviewer 4 Report
Overall, this retrograde observational study analyzing trauma-related deaths in South Korea between 2017 and 2019 offers valuable insights into preventable trauma deaths and opportunities for improvement in the care provided to trauma patients. The multi-center nature of the study, drawing data from the national database, provides a comprehensive view of the issue at hand.
It is suggested to include a brief introduction or description of the S or T codes based on the Korean Standard Classification of Diseases, seventh edition, in Section 2.1. This addition would help readers, especially those less familiar with the specific coding system, better understand the classifications used in the study.
On line 100, there is a query regarding the term 'emergency physician.' To clarify, did the authors intend to refer to emergency medicine physicians instead? Further clarity on this terminology would enhance the precision of the findings.
In conclusion, the manuscript's thorough exploration of preventable trauma deaths and its identification of areas for improvement in the three phases of care make it a highly valuable contribution to the field. With the suggested additions and clarifications, this study presents an essential resource for policymakers and healthcare professionals seeking to enhance the outcomes of trauma patients in South Korea.
Author Response
Point 1: Overall, this retrograde observational study analyzing trauma-related deaths in South Korea between 2017 and 2019 offers valuable insights into preventable trauma deaths and opportunities for improvement in the care provided to trauma patients. The multi-center nature of the study, drawing data from the national database, provides a comprehensive view of the issue at hand.
It is suggested to include a brief introduction or description of the S or T codes based on the Korean Standard Classification of Diseases, seventh edition, in Section 2.1. This addition would help readers, especially those less familiar with the specific coding system, better understand the classifications used in the study.
Response 1: Thank you for your favorable assessment of our study's findings. Following the reviewer's suggestion, we have included the following additional explanation in the 'Materials and Methods' section at line 61.
KCD is a systematic classification of diseases and deaths in Korea. It has been in use since 1952 to standardize criteria for compiling statistics on public health and medical phenomena. Based on the International Statistical Classification of Diseases and Related Health Problems (ICD), KCD provides a standardized framework for data analysis. Within KCD system, the disease classification codes S and T specifically identify health problems resulting from trauma.
Point 2: On line 100, there is a query regarding the term 'emergency physician.' To clarify, did the authors intend to refer to emergency medicine physicians instead? Further clarity on this terminology would enhance the precision of the findings.
Response 2: In response to the reviewer's feedback, the term has been revised to 'emergency medicine physician.' We thank the reviewer for their advice to ensure clarity of the terminology.
Round 2
Reviewer 1 Report
I appreciate the authors for their time and efforts to revise the manuscript and address the comments.